# Spreading of Impacting Water Droplet on Surface with Fixed Microstructure and Different Wetting from Superhydrophilicity to Superhydrophobicity

Sergey Starinskiy [1,2,*] , Elena Starinskaya [1,2] , Nikolay Miskiv [1,2] , Alexey Rodionov [1,2] , Fedor Ronshin [1,2] , Alexey Safonov [1] , Ming-Kai Lei [3] and Vladimir Terekhov [1,2]

1  S.S. Kutateladze Institute of Thermophysics SB RAS, Lavrentyev Ave. 1, 630090 Novosibirsk, Russia
2  Novosibirsk State University, Pirogova Str. 2, 630090 Novosibirsk, Russia
3  School of Materials Science and Engineering, Dalian University of Technology, Dalian 116024, China
*  Correspondence: starikhbz@mail.ru

**Abstract:** The spreading of the water droplets falling on surfaces with a contact angle from 0 to 160° was investigated in this work. Superhydrophilicity of the surface is achieved by laser treatment, and hydrophobization is then achieved by applying a fluoropolymer coating of different thicknesses. The chosen approach makes it possible to obtain surfaces with different wettability, but with the same morphology. The parameter $t^*$ corresponding to the time when the capillary wave reaches the droplet apex is established. It is shown that for earlier time moments, the droplet height change does not depend on the type of used substrate. A comparison with the data of other authors is made and it is shown that the motion of the contact line on the surface weakly depends on the type of the used structure if its characteristic size is less than 10 μm.

**Keywords:** droplet impact; spreading; superhydrophobicity; superhydrophilicity; wettability; water droplet; laser ablation; HW CVD

## 1. Introduction

The development of surface treatment methods at the micro- and nanoscale opens the way to the evolution of biomechanical technologies. Particular attention is paid to the issue of reproducing the hierarchical topologies observed in nature, possessing the principle of self-cleaning (lotus effect), retention of liquid droplets (rose petal effect), and reduction in hydrodynamic resistance (shark skin) [1–3]. The experience accumulated by researchers formed the basis of several approaches to the creation of materials called superhydrophobic, superhydrophilic, and biphilic [4–8].

The actual changes in the wettability properties are caused by two factors—the chemical composition of the surface and its topology. Depending on their combination, according to the classical studies of Wenzel [9] and Cassie-Baxter [10], surface wettability can be described in two modes. The first one (Wenzel mode) implies that the liquid is in full contact with the surface. In this case, the topology development entails increasing hydrophobicity for hydrophobic materials and hydrophilicity for hydrophilic ones by the expression $cos\,\theta_r = r{\cdot}cos\,\theta$, where $r$ is the roughness value, $\theta$ is the contact angle (CA) of the smooth surface, and $\theta_r$ is the contact angle of the rough surface. Thus, the initially hydrophilic silicon surface acquires superhydrophilic properties after texture creation [11]. In the second mode (Cassie–Baxter), an air layer, called a plastron, is trapped in the cavities formed by the microtexture. It is the presence of air pockets that ensures the enhancement of hydrophobic properties up to reaching the superhydrophobic state. The static contact angle can be estimated by the following expression: $cos\theta_r = \sum_n f_n cos\theta_n$, where $f_n$ is the total area of each interface under the droplet per unit projected area. These simple models are very illustrative and have good applicability; however, they have some limitations [12–14],

for example, in describing the rose petal effect when a superhydrophobic state with high adhesive strength is realized on a hierarchical micro/nanostructure [15].

Materials with both superhydrophilic [16,17], superhydrophobic [18,19], and biphilic [20] properties are very promising in terms of passive (energy-free) control of the liquid droplets' interaction with a solid wall. This affects important applications such as spray cooling [21], fuel combustion [22], additive technology [23], coating [24], deicing [25], biofouling, and surface contamination [26]. One of the actual challenges is the study of multiphase phenomena [27,28], in particular, the behavior of liquid droplets [29] including interphase phenomena. However, there is currently no unequivocal understanding of the material wettability effect on the dynamics of the falling droplet spreading on the surface. Thus, the data available in the literature on the dynamics of the falling droplets spreading on superhydrophobic surfaces with similar contact angles under close conditions can differ appreciably [30–32]. According to the results of Pachchigar et al. [33], maximum droplet spreading on a structured fluoropolymer with contact angles of 105–154° is independent of surface morphology for Weber numbers $We = \frac{\rho D_0 v^2}{\sigma} < 40$, where $\rho$ is the density, $D_0$ is the droplet diameter, $v$ is the droplet velocity, and $\sigma$ is the surface tension. On the other hand, Pan et al. [34] observed a significant difference in the dispersion of the falling droplets, although they used materials with a contact angle range (77–145°) close to the work of Pachchigar et al. [33]. Moreover, a difference in the spreading dynamics was found in the investigation of Lv et al. [35], where the influence of nanostructure on the droplet bounce dynamics from surfaces with close roughness at the microlevel was clearly shown. This agrees with the results of other authors [36,37], which show that for the same static contact angles, a droplet on the surface can be in both the lotus (without pinning) and rose petal (with pinning) states, which affect the liquid spreading on the surface. These results are in agreement with the MDPD calculations carried out by Du et al. [38], where it is shown that sufficiently high contact angles (~145°) can be achieved in both the Wenzel and Cassie–Baxter modes.

Despite the apparent simplicity, the fall of droplets on a solid wall is a complex two-phase process. Upon primary contact with the wall at the liquid–gas interface, capillary-surface waves are generated, which leads to the formation of pyramidal structures [39] with a characteristic wavelength $\sigma/\rho V$, where $\rho$ is density, $\sigma$ is surface tension, and $V$ is velocity. Further movement of the liquid leads to the formation of a dish-like or torus-shaped topology. According to Renardy et al. [39], surface dry-out is determined by the ratio $We = 1590/Re^{1.49} + 3.62$. The formation of a dry cavity leads to the capture of an air bubble during the retraction stage. The higher the falling velocity, the higher the side lamella velocity. To describe the maximum spreading $\beta_{max}$, various analytical models are used, which, as a rule, do not take into account the surface characteristics of the wall [40]. However, many authors have noticed the incorrectness of this approach for low $We$ numbers in conditions before splashing. Thus, it was shown by Ukiwe and Kwok [41] that the experimental results are much better described by taking into account the contact angle. For modes involving the development of instability and subsequent splashing, as a rule, the boundary between deposition and splashing is determined by the parameter $K = We^{0.5} Re^{0.25}$. However, according to Roisman et al. [42], even in this case, it is necessary to take into account the morphology of the analytical surface, and it was proposed to redefine the Reynolds number, taking into account the surface roughness. After reaching the maximum spreading, a reverse flow occurs, which cannot be independent of the receding contact angle. Much attention was paid to this issue in the work of Wang and Fang [32], and quite complex analytical approaches were proposed for constructing retraction curves. However, the authors took into account only the contact angles, but not the topology of the surface.

The influence of surface topological characteristics on the falling droplet process continues to be intensively studied as there are now reliable ways to control surface roughness [2,43] or to create periodic structures with different spatial distributions [30,44–47]. A noticeably lower amount of work is devoted to the study of the dynamics of liquid

spreading over surfaces with different wettability but the same morphology, except the results obtained using different liquids [43,48–51]. Zhang et al. [52] considered the effect of surface wettability on the water droplet spreading in a wide range of Weber numbers We = 0–3000 (although the data in the work start from We = 80). Hydrophilic, hydrophobic, and superhydrophobic surfaces were investigated with CA = 30–150°. Samples were obtained by plasma treatment and functionalization methods with hydrophobic agents. According to the proposed mechanism, the wettability property mainly affects the rise in the lamella edge and the subsequent air leakage, which has a significant influence on droplet spreading and splashing characteristics. Sun et al. [53] examined falling droplets on surfaces with CA = 5 and 134°. Different wettability was achieved by UV exposure to titanium oxide, suggesting that the surface morphology remains unchanged. It was shown that for superhydrophilic surfaces, the spreading dynamics also depend on the material texture, which determines the spreading velocity; in particular, the detachment of secondary droplets is possible only in the case when the spreading velocity is above the falling droplet velocity.

Further analysis of the effect of transitions between Cassie–Baxter and Wenzel modes is required as several authors have shown a very pronounced effect on the velocity of liquid movement along the wall [45,54], even though more and more attention has recently been paid in the literature to the issue of a water droplet falling on surfaces with the rose petal effect [2,55,56]. Most papers consider the effect of simultaneous changes in surface structure and wettability, while these parameters have significantly different effects on the falling liquid droplet spreading. However, there is still no complete understanding of the influence of the surface topology and wettability on the dynamics of falling droplet spreading, and the data available in the literature are quite scattered. There are no data on the flow dynamics for materials with structures similar to rose petals but different contact angles. Thus, the purpose of this work is to fill the research gap and investigate the surface wettability effect on the liquid droplet spreading on surfaces with the same morphology. For the first time, we studied water droplets falling on surfaces with a contact angle from <5° (superhydrophilic) to 155° (superhydrophobic) with a fixed surface topology close in structure to a rose petal. Experiments were performed in a wide range of Weber numbers We = 0.3–33 for different droplet sizes in the range of $D_0$ = 2–3 mm. For a detailed study of the spreading dynamics, simulations were performed using the lattice Boltzmann method, which allows us to analyze the velocity field in time.

## 2. Experimental Setup

In this work, the change in the surface wettability was achieved in a two-stage process. During the first stage, the surface was irradiated in the air by pulses of the basic harmonic Nd:YAG laser (home-made) with a wavelength of 1064 nm and a pulse duration of 11 ns. The average energy density in the beam was 3.6 J/cm$^2$; thus, conditions favorable for the formation of a special hierarchical structure of the laser spot were created on the surface [11]. The material was irradiated in the mode of beam scanning along the surface with an area of 12 × 18 mm. The laser spot was 0.4 mm$^2$, the number of laser pulses per spot was about 60, and the total number of surface preparations was 30,000. The overlapping of laser spots was 60% and controlled by laser pulse frequency and scanning velocity. The initial surface of monocrystalline silicon with natural oxidation had a contact angle of ~55°. Laser processing of the silicon surface led to the formation of a self-organized periodic structure consisting of alternating hillocks and hollows with a characteristic spatial size of about 10 μm. In addition to the micron-sized periodic structure, there was a second level of nanometer roughness or porosity formed by ablation products returning to the surface. The contact angle on the laser-processed surface was less than 5°, i.e., the substrate became superhydrophilic. The texture can retain its properties for a very long time under various external influences, particularly during pool boiling [57].

In the second stage, the laser-treated samples were hydrophobized by applying a fluoropolymer coating of different thicknesses by the Hot Wire Chemical Vapor Deposition

(HW CVD) method [58]. In the HWCVD method, a hot catalytic metal wire mesh is used to activate the precursor gas. The experimental setup for depositing coatings was described in detail by Safonov et al. [58]. Hexafluoropropylene oxide $C_3F_6O$ was used as the precursor gas of the fluoropolymer film. Silicon substrates were placed in a cooled substrate holder in a vacuum chamber. At a distance of 30 mm above the substrates, there was a catalytic activator in the form of a mesh made of a 0.5 mm diameter helically coiled nichrome wire with a spacing of 20 mm. The mesh temperature was fixed at 580 °C and monitored by volt-ampere measurements (Mastech MS8050, Huayi Mastach Co., Shenzhen, China). The precursor gas pressure in the deposition chamber was 0.5 Torr and the gas flow rate was 20 sccm. The substrate temperature was about 30 °C during deposition. The thickness of the fluoropolymer coating was controlled by varying the deposition time within the range from 30 s to 750 s. The fluoropolymer coating does not affect the material topology at both micro- and nanolevels [59]. The thickness of the fluoropolymer coating did not exceed 50 nm and was controlled by the deposition duration. This technique already allows us to achieve stable superhydrophobic properties with a contact angle greater than 160° [59]. The microstructures of the obtained samples and the rose petal were very similar to each other (Figure 1).

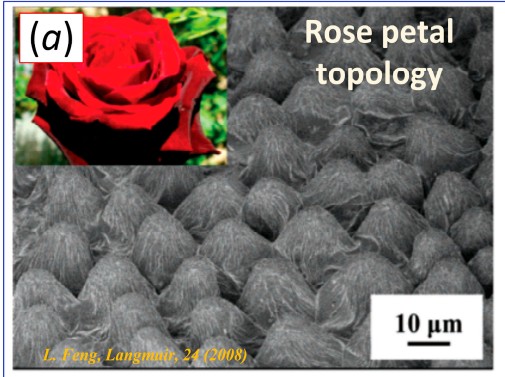 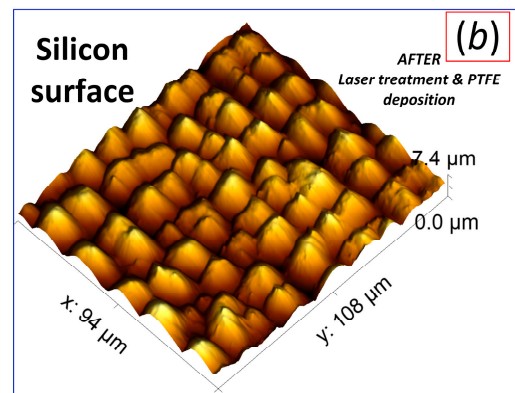

**Figure 1.** Comparison of the surface topology of the rose petal (**a**) (adapted from [15]) and samples synthesized in the present work (**b**).

Both advancing contact angle (ACA) and receding contact angle (RCA) were measured using the sessile drop method on a KRUSS DSA 100 (KRUSS GmbH, Hamburg, Germany). The liquid droplet is placed on the substrate using an automatic dosing system. In the first stage, the advancing contact angle is measured when liquid is pumped into the droplet. At the initial moment, the contact angle increases as the contact line is pinned. Then, the depinning of the contact line takes place, the contact angle becomes constant, and the measurements are taken at this moment. In the second stage, the receding contact angle is measured in a similar way as the liquid droplet is reduced in volume. The results are presented in Table 1. The obtained contact angles make it possible to fully characterize the surface wetting hysteresis.

**Table 1.** Wettability of samples as a function of fluoropolymer layer thickness (PTFE).

| Sample | ACA, ° | RCA, ° | PTFE Thickness, nm |
|---|---|---|---|
| Surface 1 | <5 | <5 | 0 |
| Surface 2 | 22 | 5 | 2 |
| Surface 3 | 50 | 17 | 5 |
| Surface 4 | 90 | 20 | 12 |
| Surface 5 | 145 | 22 | 25 |
| Surface 6 | 160 | 159 | 50 |

A scheme of the experimental setup to study the dynamics of the spreading of Milli-Q water droplets falling on the synthesized surfaces is shown in Figure 2a. Visualization was performed with a Phantom VEO710 high-speed camera (Vision Research, Inc. Wayne, NJ, USA). Typical images of the process are shown in Figure 2b. The main analyzed parameters were droplet height $H$ and contact line diameter $D$ (Figure 2b). Occasionally, during the spreading of the droplet, the central part was hidden behind the droplet side parts, in which case $H$ was measured as the maximum height of the lamella. The experiments were performed for distilled water droplets 2–3 mm in size falling from a height of 1–60 mm (Weber numbers We = 0–33 and Reynolds numbers Re = 250–3000), which corresponds to deposition, spreading, partial rebound, and rebound dependent on wettability surface regimes [31,50]. In other words, we worked under no splash conditions. The Bond number Bo ~ 1, i.e., the dynamics of droplet spreading was predominantly determined by surface tension.

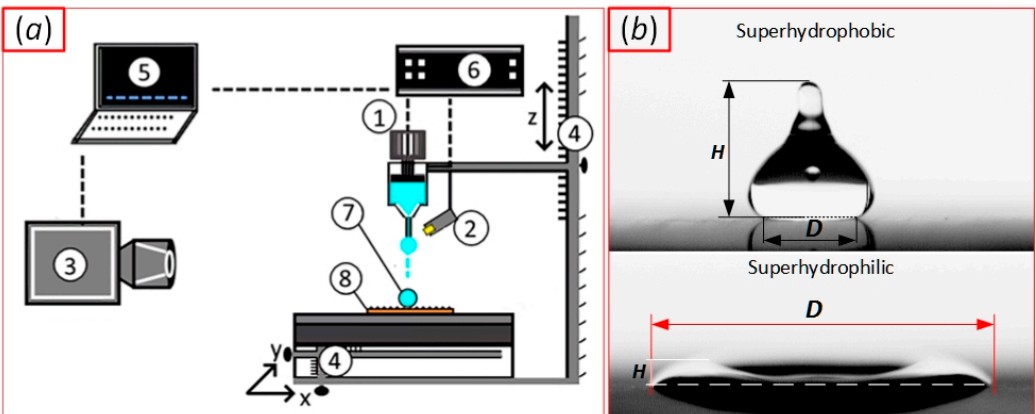

**Figure 2.** (**a**) Experimental setup for measuring the droplet impact and spreading characteristics on the substrate surface: 1—injection pump, 2—droplet break-away sensor, 3—high-speed video camera Phantom VEO710, 4—3-axis positioning table, 5—computer, 6—microcontroller unit, 7—droplet, 8—substrate. (**b**) Typical snapshots of droplet spreading on superhydrophilic and superhydrophobic materials.

### 3. LBM Simulation

In this work, the simulation was performed using the multiple-relaxation-time lattice Boltzmann method (MRT-LBM) [60]. LBM is currently an effective method for studying fluid flows; its advantages are most clearly seen in modeling multiphase flows and surface phenomena. There are several approaches to describe the interphase interactions in the LBM; in this work, we chose the pseudopotential model [61,62], in which the interparticle interactions are represented as a force based on the potential, which depends on the density of the medium. When modeling in the present work, the medium was represented as a one-component medium with separation into liquid and vapor phases. The conditions under which the equilibrium properties of vapor and liquid are close to the properties of air and water in the experiment were determined by the given temperature.

It is important to note that in this work, we used a two-dimensional model, simplifying the problem significantly. However, there is a successful experience of the 2D approach to solving the problem of droplet surface interaction dynamics [63–66]. In addition, the goal of modeling is to obtain qualitative data on the velocity distribution, which was difficult to obtain in the experiment. Thus, we expect to provide important information for understanding the physical mechanism of the interaction of a drop with a surface despite the accepted 2D simplification.

For a simple description of the physical phenomena considered in this work, we used a two-dimensional formulation, which, despite the obvious limitations, allows us to

qualitatively describe the process of droplet spreading [63]. The basic LBM equation, which uses a multiple-relaxation-time collision operator, is as follows:

$$f_\alpha(x + e_\alpha, t + \Delta t) = f_\alpha(x, t) - \overline{L}_{\alpha\beta}\left(f_\beta - f_\beta^{eq}\right)\Big]_{(x,t)} + \Delta t\left(S_\alpha - \frac{1}{2}\overline{L}_{\alpha\beta}S_\beta\right)\Big]_{(x,t)} \tag{1}$$

where $f_\alpha(x, t)$ is the density distribution function; index *eq* is its equilibrium value; $t$ is time, $x$ is a coordinate, $e_\alpha$ is a discrete set of velocity vectors in the $\alpha$ direction; $S_\alpha$ is the term describing the action of various forces, $\overline{L} = M^{-1}LM$ is the collision matrix, in which $M$ is the orthogonal transformation matrix and $L$ is the diagonal matrix. Using the above transformation matrix, the distribution function is transformed to the moment space $m = Mf$ and $m^{eq} = Mf^{eq}$.

For the D2Q9 lattice used in this paper, the matrices $M$ and $L$ and vectors $e_\alpha$ and $f^{eq}$ are given by Bouzidi et al. [67]. The solution of Equation (1) is carried out in two stages. The first stage is the transition to the space of moments and carrying out the process of collisions.

$$m^* = m - L(m - m^{eq}) + \Delta t\left(I - \frac{L}{2}\right)S \tag{2}$$

The second stage is to return to the distribution function $f^* = M^{-1}m^*$ and conduct the distribution process:

$$f_\alpha(x + e_\alpha, t + \Delta t) = f_\alpha^*(x, t). \tag{3}$$

After that, the macroscopic quantities are defined as $\rho = \sum_\alpha f_\alpha$ and $\rho V = \sum_\alpha e_\alpha f_\alpha$. Phase separation was modeled according to the pseudopotential approach. According to this model, a force acts in a gas or liquid [62]:

$$F = -G\psi(x)\sum_\alpha w_\alpha \psi(x + e_\alpha)e_\alpha, \tag{4}$$

where $G$ is the interaction force, $\psi(x)$ is the potential, and $w_\alpha$ is the weighting factor in the $\alpha$ direction. The interaction potential was determined according to [62]

$$\psi(\rho) = \sqrt{\frac{2(p_{EOS} - \rho c_s^2)}{Gc^2}}, \tag{5}$$

where the pressure is determined from the state equation, in our case, Carnahan–Starling:

$$p_{EOS} = \varrho RT\frac{1 + \frac{b\rho}{4} + \left(\frac{b\rho}{4}\right)^2 - \left(\frac{b\rho}{4}\right)^3}{\left(1 - \frac{b\rho}{4}\right)^3} - a\rho^2, \tag{6}$$

where parameters $a$ and $b$ are functions of critical temperature and pressure, respectively. Following the approach of Li et al. [64], we chose $R = 1$, $b = 4$, and $a = 0.25$. The introduction of force (4) into Equation (2) was carried out according to the exact difference method [68], and gravity was included in the same way. Such an approach to the integration of interfacial interaction significantly suppresses non-physical currents arising in the case of the original method [61,62] due to the non-isotropy of discrete operators. It allows us to achieve liquid-to-gas density ratios of ~1000, i.e., to simulate a water/air system at normal conditions.

Aspects of the construction of boundary conditions in the LBM for multiphase applications using the pseudopotential approach were studied in detail by Khajepor [69]. The top and bottom boundaries of the domain were considered to be solid surfaces, while the periodic boundary condition was applied to the side boundaries.

The interaction of a fluid with a solid surface was modeled by a similar approach:

$$\boldsymbol{F} = -G\psi(\boldsymbol{x})\sum_\alpha w_\alpha \psi(\varrho_w)H(\boldsymbol{x} + e_\alpha)e_\alpha, \tag{7}$$

where $H$ is a function taking the value $H = 1$ at the points of the solid surface and $H = 0$ at any other points. $\varrho_w$ is a parameter characterizing the wetting degree and determining the value of the contact angle. Note that, unlike most works using such an approach, in the present work, the parameter characterizing the wetting degree $\varrho_w$ was not a constant value both in space and time. As shown below, the surfaces considered in the work have a significant contact angle hysteresis, so the value of the contact angle changes significantly when the liquid flows over the surface and during the reverse process. Thus, the initial state $\varrho_w$ was taken from the considerations of reproducing the advancing contact angle according to the experiment and then, depending on some "critical" pressure on the wall, "switched" to the receding contact angle.

In all calculations, the distribution corresponding to a circular droplet of a given diameter located near the surface and having a given velocity was set as the initial condition. The boundary conditions on the upper and bottom surfaces were assumed to be solid surfaces with a given $\varrho_w$, and the side faces were assumed to be periodic. Preliminary calculations were performed to determine the dependence of the contact angle value on $\varrho_w$, the dependences of vapor/liquid density and surface tension on temperature, and to verify mesh convergence. For most cases, a grid of $768 \times 768$ cells was used, while, in lattice units, the drop diameter was 164 cells, the relaxation time was 0.515, and the initial drop velocity was 0.021. Especially for the superhydrophilic surface, the horizontal resolution of the grid was increased up to 2048 unity.

## 4. Results and Discussion

Figure 3a shows images of a 2.3 mm diameter water droplet falling on surfaces with different wettability at a velocity of 0.3 m/s at the moment of the collision, which corresponds to the dimensionless criteria of $We = 3$, $Re = 775$, and $Bo = 0.73$. Based on the sweep of the droplet falling dynamics, several stages can be distinguished. The first one is the inertial stage with a duration of less than 3 ms. Thus, a capillary wave is formed upon droplet contact with the surface, which propagates across the droplet surface and deforms the droplet into a pyramidal structure [39]. At this stage, the shape of the droplet is determined only by the Weber number (falling velocity) and weakly depends on the type of surface used. The only exception is a superhydrophilic surface for which, at small Weber numbers ($We < 3$), the contact line velocity may exceed the lateral spreading velocity of the droplet. However, this has an insignificant effect on the dynamics of the droplet height $H$.

The inertial stage is followed by the viscous spreading stage until the maximum lateral droplet size is reached. For all the samples studied except for the superhydrophilic one, the droplet behavior is qualitatively almost identical. For superhydrophilicity, we see the gradual spreading of liquid. For other cases, we observe the formation of a toroidal structure, i.e., the central part of the droplet is lower relative to its sides. The influence of surface wettability can be seen in the dynamic contact angle (Figure 3, $t = 4.6$ ms). For the superhydrophilic case at the beginning of the viscous flow stage, the achievement of a local maximum on the dependence $H(t)$ is observed. This process is based on the same mechanism as the droplet emission induced by ultrafast spreading on a superhydrophilic surface [53]. Like Sun et al. [53], we observe droplet emission from the surface of a superhydrophilic material at small Weber numbers ($We < 0.1$). The viscous stage ends when the droplet on the substrate reaches its maximum lateral size, which is determined by the size of the region where the droplet is pinned to the surface. Next, the retraction stage is realized. For hydrophilic surfaces, the dynamics of liquid motion at this stage are qualitatively similar, although the difference in the dynamic contact angle is already more pronounced (Figure 3, $t = 7.2$ ms). For the hydrophobic surface, the return flow coincides for some time with the flow for superhydrophobic material. However, while, for the superhydrophobic surface, a decrease in the contact line up to the detachment is observed, for the hydrophobic one, there is a pinning of the droplet to the surface. After the contact line pinning on hydrophilic surfaces, the contact line diameter changes weakly, and complete relaxation occurs significantly later, after at least 500 ms. At this stage,

gradually decaying oscillations can be seen for the droplet height value. The frequency of oscillations is determined by the contact angle, although the general behavior of the droplets is almost identical. Attenuation is slowest on a hydrophobic surface with high adhesive properties. After attenuation, the contact angle becomes close to 145°, as was recorded in the measurements on the KRUSS DSA 100. The observed nonmonotonicity in the frequency of oscillations depending on the contact angle is one of the directions for further research.

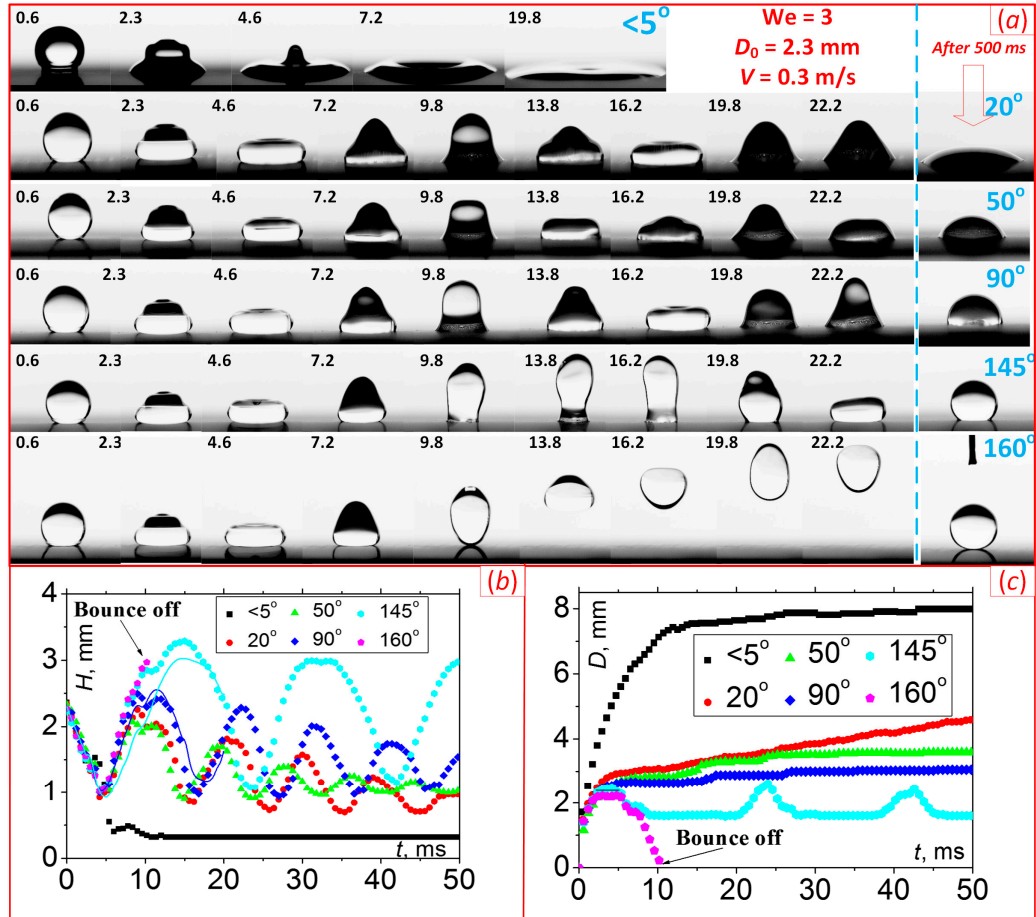

**Figure 3.** (**a**) Snapshots of droplet $D_0$ = 2.3 mm falling on surface with different contact angles for *We* = 3, numbers are time in ms; the dotted line separates the snapshots after 500 ms (**b**) Variation in droplet height and (**c**) contact line diameter with time. Contact angle θ < 5° (■), 20° (●), 50° (▲), 90° (◆), 145° (●), 155° (●). Solid lines in (**b**) are LBM modeling for surface 4 and 5.

We do not observe any fundamental difference in droplet behavior depending on the size, as can be seen from Figure 4, which compares the change in droplet height dynamics with time for three sizes. Time *t\**, at which inertial flow is realized, is found—for this stage *t* < *t\**, the behavior of the change in droplet height with time for superhydrophilic and superhydrophobic materials is the same (as well as for intermediate contact angles). Moreover, values of *t\** do not depend on droplet falling velocity and are determined only by droplet size at least for *We* < 30 (Figure 4b). This point allows the comparison of data with other authors' data and calculated result verification for different droplet sizes. It is expected that for the dependence of *t\** on $D_0$, the power is 3/2 as the inertial-capillary number $t_c = \sqrt{\frac{\rho R_0^3}{\sigma}}$ (Figure 5a) [31]. This expression is obtained in the Hertz problem on the deformation of the ball against the surface [70]. This moment corresponds to reaching the maximum droplet spreading or the minimum droplet height. We cannot register exactly the moment of achieving the minimum droplet height center in the experiment, because it

is closed by the lamella (see Figure 3, $t$ = 4.6 ms), which is why in Figure 4a, we observe a plateau in the area of transition from viscous spreading to return flow at the stage of $t$ = 4–5 ms (for $D_0$ = 2.3 mm). The dependence of the time moment of droplet rebound from the superhydrophobic surface $t_b$ on $D_0$ also has the same power of 3/2 [70] (Figure 5).

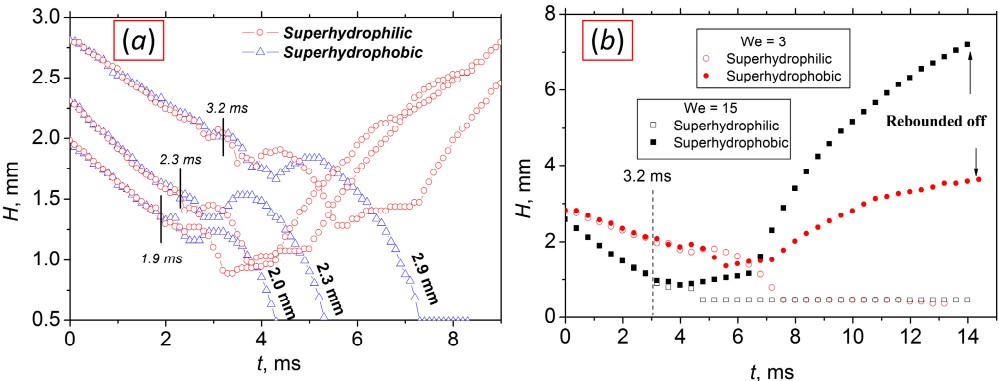

**Figure 4.** (**a**) Evolutions of height of droplets with different diameters impacted on superhydrophobic and superhydrophilic surfaces We = 3. (**b**) Evolutions of height of droplets for We = 3 and 15 $D_0$ = 2.9 mm.

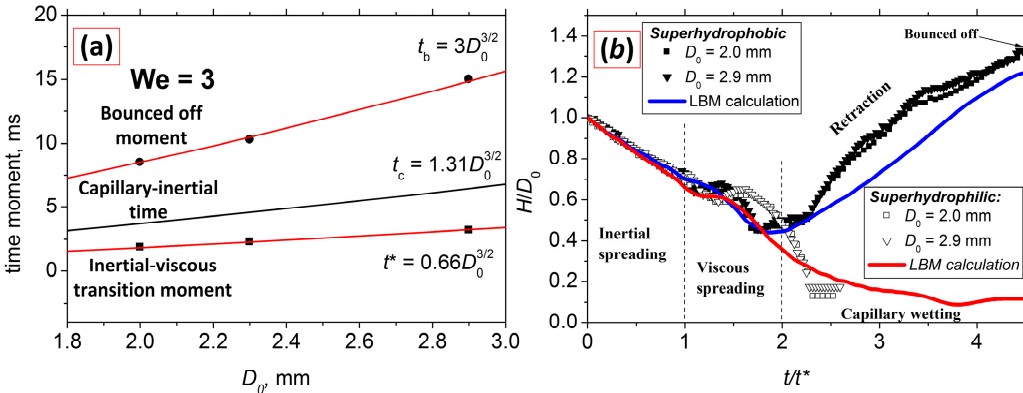

**Figure 5.** (**a**) Dependence of time $t^*$, capillary-inertial time $t_c$, and droplet rebound moment $t_b$ on the diameter. (**b**) Comparison of experimental results for $D_0$ = 2 and 2.9 mm in relative coordinates, $We$ = 3.

Further, we use the value of $t^*$ to generalize the obtained data and compare it with the other authors. A similar approach was used for time $t_c$ [71]. For superhydrophilic surfaces, a local minimum (for $We$ < 10) or a transition to the plateau ($We$ > 10) is fixed at this time, while time $t_c$ and $t_b$ are weakly applicable. In particular, as it has been noted above, for small Weber numbers ($We$ < 3) at sufficiently high spreading velocities (exceeding the droplet velocity) in the region $t$ < $t_c$, a local increase in droplet height is observed [72] and may lead to detachment of secondary droplets, which we have recorded as with Sun et al. [53] at We ~ 0.1.

Figure 6 shows a direct comparison of the water droplet dynamics calculation for a diameter of 2.9 mm on the superhydrophilic and superhydrophobic surfaces with the experimental results. A qualitative and quantitative agreement is observed (Figure 5b). In the calculation, as well as in the experiment, there is a local maximum associated with the limited lateral velocity of the droplet spreading, both on the superhydrophobic and superhydrophilic surfaces. The experimental and calculated times of the liquid droplet rebound from the superhydrophobic surface are also in good agreement. We associate some discrepancies that we observe at the viscous spreading and retraction stages with the limited simulation capabilities for the case of 2D simulation. Figure 5b successfully shows

the idea of introducing the time $t^*$. This is the time, observed for the droplet height, during which the wave reaches the surface. Based on the time value thus determined, it is possible to generalize the data without additional information about the properties of the liquid used or the initial size of the droplets.

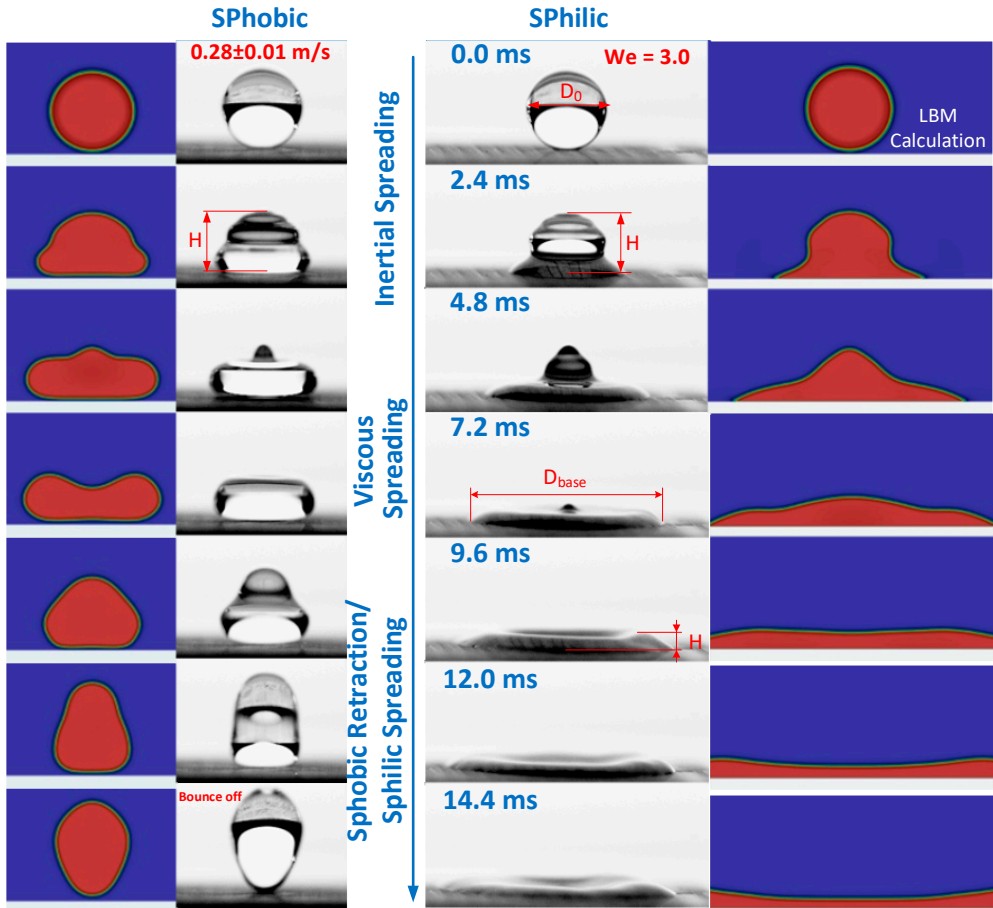

**Figure 6.** Comparison of the calculated and experimental dynamics of the water droplet interaction ($D_0 = 2.9$ mm) with superhydrophobic and superhydrophilic surfaces. The colors shows qualitatively different densities to recognize the interface.

From Figures 5b and 6, one can also trace all the characteristic stages of the process of interaction of a droplet with a surface both in the superhydrophilic and superhydrophobic cases, as well as compare them with each other. The first stage is inertial, limited by the propagation time of the surface wave to the droplet apex. It can be seen that the upper part of the droplet on fundamentally different surfaces remains similar, while the droplet height is the same for superhydrophobic and superhydrophilic surfaces (Figures 6 and 7). In the second stage, the viscous one, a rapid (especially in the hydrophilic case) spreading of the droplet along the surface is observed. Finally, the final stage for superhydrophobic and superhydrophilic surfaces is fundamentally different. In the first case, the spreading is replaced by runoff and droplet rebound from the surface; in the second case, the droplet completely spreads over the surface.

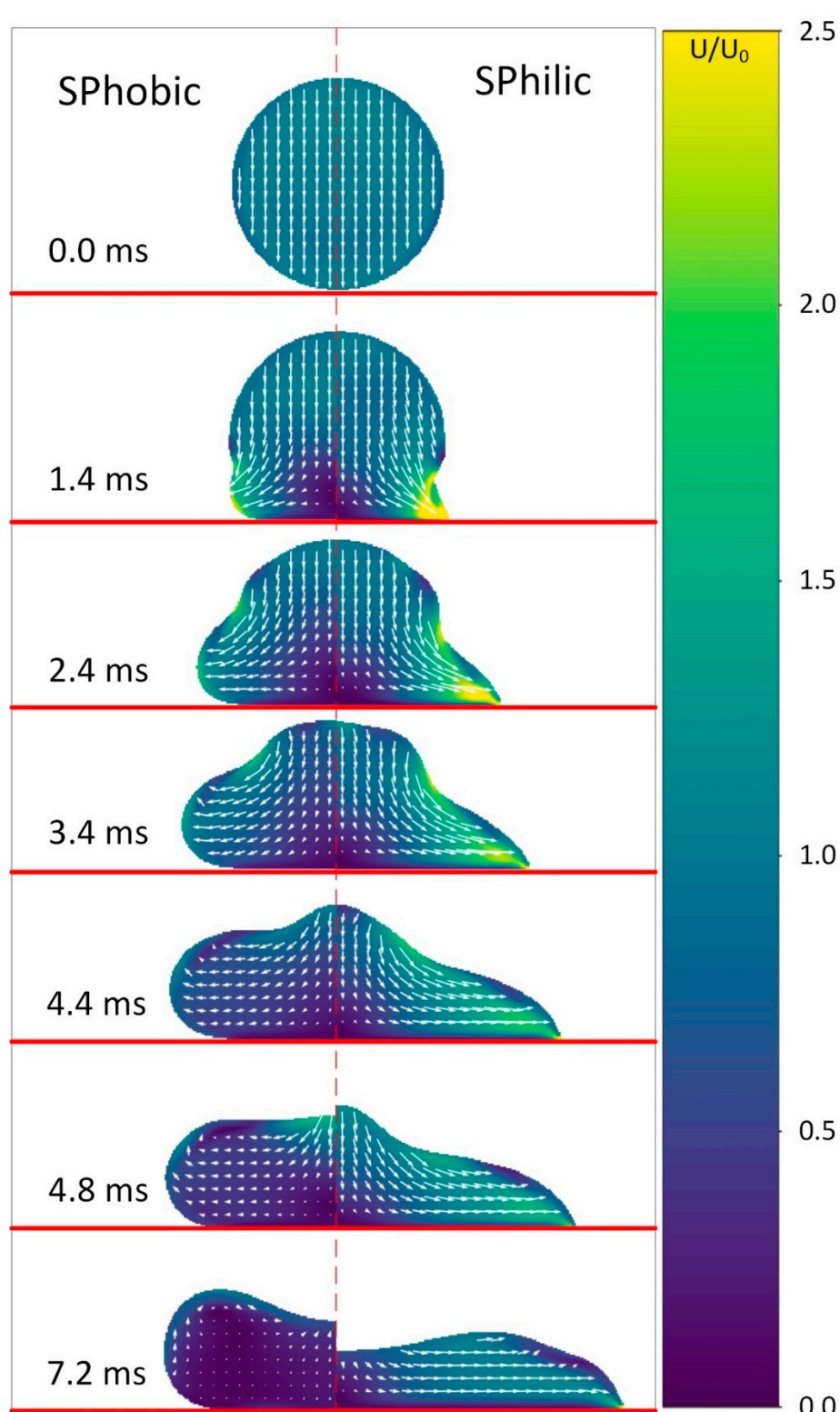

**Figure 7.** Velocity field in the droplet with $D_0$ = 2.9 mm falling on the superhydrophobic and superhydrophilic surfaces at $We$ = 3.

The good agreement between calculation and experiment for the inertial spreading stage allows us to analyze in detail the dynamics of liquid flow (see also Figure 3b for surfaces 4 and 5). Thus, Figure 7 shows the velocity field in a droplet in contact with surfaces with different wettability. It can be seen that wave motion on the interface appears at the moment when the droplet touches the surface and reaches the droplet apex at $t^*$, which is

the physical meaning of this parameter. It is seen that for $t = 3.4$ ms $> t^* = 3.2$ ms, there is a divergence in droplet height associated with wave deformation of the interface. Further spreading over the superhydrophobic and superhydrophilic surface becomes perfectly different throughout the volume of the droplet. By the time $t_c$, the velocity of the droplet on the superhydrophobic surface converges to zero at all points except the interface. However, for superhydrophilic surfaces, $t_c$ has no physical meaning.

Using the $t^*$ parameter, we compare the dynamics of falling droplets spreading on different surfaces with the data available in the literature (Figure 8). The list of references and experimental conditions are given in Table 2.

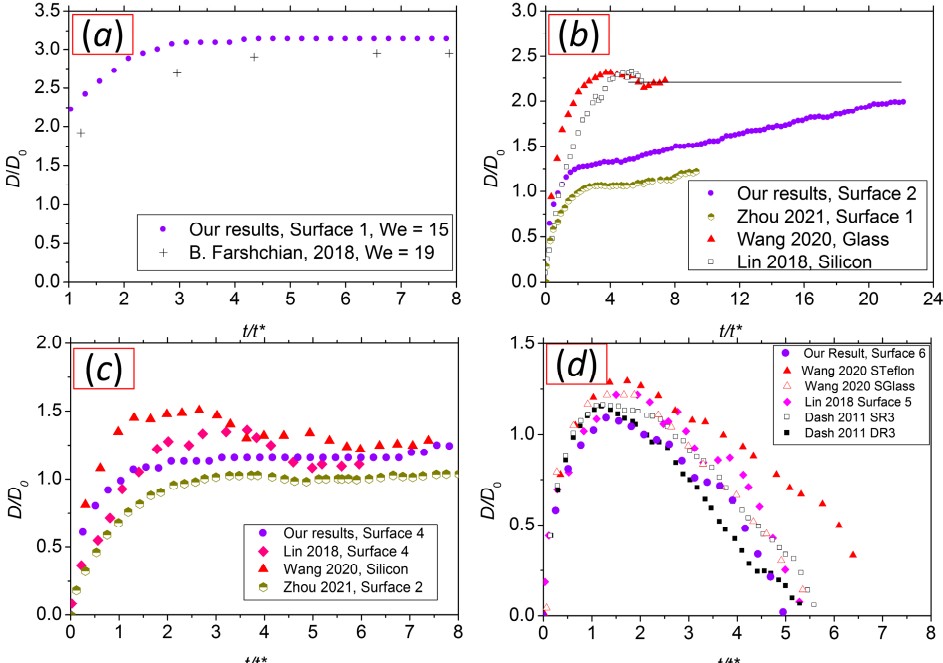

**Figure 8.** Comparison of experimental data on the dynamics of water droplets spreading on the (**a**) Superhydrophilic surfaces at $We = 15$–19, (**b**) surfaces with $CA = 20°$–40° at $We = 1.5$–4, (**c**) surfaces with $CA = 84°$–110°, $We = 1.5$–4, (**d**) superhydrophobic surfaces $CA = 140°$–160°, $We = 2$–4.

We were unable to find data on the water droplets spreading dynamics on superhydrophilic surfaces at $We = 3$ in the literature; however, comparison with the data of Farshchian et al. [73] shows good agreement at $We \sim 17$ (Figure 8a). Figure 8b shows a comparison of contact line dynamics for droplets falling at $We = 1.5$–3 on our textured surface (Surface 2) with a contact angle of 20°, a surface with a carbon nanotube forest with a contact angle of 29° [74] and smooth silicon and glass surfaces from other works [31,32]. The droplet spreading on the textured surfaces is identical; the difference in the maximum droplet spreading can be explained by a small difference in the contact angles and the $We$ number. At the same time, there is a significant difference in the contact line motion on smooth and textured surfaces at the stage of viscous spreading ($t/t^* > 1$): first, there are no oscillations of the contact line, due to liquid flowing into the texture; second, reaching the equilibrium state requires considerable time, which may be related to the limited rate of water penetration into the material structure. For high contact angles (i.e., for thick fluoropolymer coating), lateral droplet movement over textured surfaces is limited to the pinning region reached by the end of the inertial spreading stage (Figure 8c), while oscillations occur on the smooth surface. However, the contact line diameters are close to each other for all surfaces considered at $t/t^* > 6$. The data for falling droplets on superhydrophobic surfaces are qualitatively similar (Figure 8d). The maximum droplet spreading and detachment distances from the surface at approximately the same time are reached under the condition that it is wetted in the Cassie–Baxter state. It was shown by Wang and

Fang [32] that the transition between Cassie–Baxter and Wenzel wetting states changes the dynamics of liquid motion at the retraction stage (STeflon Figure 8d), which can lead to later droplet detachment and partial adhesion of liquid to the surface.

**Table 2.** References for comparing the dynamics of water droplet spreading on surfaces with different wettability.

| Reference | We | $D_0$, mm | Surface | ACA, ° | RCA, ° | SCA, ° |
|---|---|---|---|---|---|---|
| S. Dash, et al. 2011, [30] | 2.8 | 2.2 | Single-roughness surface (SR3) | 155 | 122 | 144 |
| S. Dash, et al. 2011, [30] | 2.8 | 2.2 | Double-roughness surface (SR3) | 165 | 155 | 166 |
| S. Lin, et al. 2018, [31] | 2 | 2.3 | Fractal-like network of hydrophobized silica shells on clean glass slides (surface 5) | 163 | 159 | 161 |
| F. Wang, et al. 2020, [32] | 4 | 2.5 | Sanding Teflon (STeflon) | 146 | 137 | - |
| F. Wang, et al. 2020, [32] | 4 | 2.5 | Superhydrophobic solution NeverWet on a piece of clean glass (SGlass) | 158 | 153 | - |
| S. Lin, et al. 2018, [31] | 2 | 2.3 | Silanized silicon wafers (surface 4) | 111 | 100 | 106 |
| F. Wang, et al. 2020, [32] | 4 17 | 2.5 | Silicon | 92 | 74 | - |
| F. Wang, et al. 2020, [32] | 4 | 2.5 | Glass | 46 | 21 | - |
| S. Lin, et al. 2018, [31] | 4 | 2.5 | Silicon | 31 | - | 27 |
| B. Farshchian, et al. 2018, [73] * | 19 | 2.3 | Plasma-treated nanoparticles on PMMA | - | - | 9 |
| W. Ding, et al. 2022, [47] ** | 10.5 | 2 | Salinized smooth microcones on silicon surface (SP8H20) | - | - | 93 |
| W. Ding, et al. 2022, [47] ** | 10.5 | 2 | Salinized smooth microcones on silicon surface (SP8H20) | - | - | 134 |
| W. Ding, et al. 2022, [47] ** | 10.5 | 2 | Salinized rough microcones on silicon surface (RP8H27) | - | - | 159 |
| M. Zhou, et al. 2021, [74] | 1.5 17.7 | 2 | Carbon nanotube forest treated by plasma (Substrate 1) | - | - | 29 |
| M. Zhou, et al. 2021, [74] | 1.5 17.7 | 2 | Carbon nanotube forest treated by plasma (Substrate 2) | - | - | 84 |
| M. Zhou, et al. 2021, [74] | 1.5 17.7 | 2 | Carbon nanotube forest treated by plasma (Substrate 3) | - | - | 147 |

Notes: * We were unable to find data to compare superhydrophilic surfaces for a droplet falling from *We* = 3, so we give an example for *We* = 19. ** In Ding et al. [47], a change in wettability was achieved either by changing the texture or the liquid. Contact angles 134° and 159° were obtained for a water–ethanol mixture of 35%, and the contact angle of 93° was obtained for a mixture of 67%.

Of particular interest is the droplet behavior the Surface 5, which, on the one hand, has high adhesion characteristics and, on the other hand, has a large value of the contact angle of ~145°. The droplet behaves on a superhydrophobic surface for a considerable time up to $t/t^*$ ~ 3.5 until the contact line pinning occurs. As a result, the droplet fails to detach from the surface. It is difficult to determine exactly at what point the droplet pinning to the surface occurs; however, unlike CA ~ 90°, the adhesion area is slightly smaller than the droplet diameter, i.e., it can be realized both at the inertial stage and later, as it was described by Lee et al. [45]. The authors of the paper report two types of transitions depending on the *We* number between the Cassie–Wenzel wetting states at different times. In the first case, liquid pinning is realized at the inertial stage. In the second case (at lower Weber numbers), wetting is observed at a stage much later in time during recoil and before the rebound. In addition, the rebounding droplet appears pinned to the surface through a small and central wetted area. We still suggest that fixation (i.e., the Cassie–Wenzel transition) occurs at the first stage of contact of the droplet with the surface. This is supported by the absence of dependence on the fixation moment on the *We*. The fixing moment for Surface 5 for the droplet diameter of 2.3 mm is reached at $t$ ~ 8 ms (see

Supplementary information). In addition, the spreading data for Surfaces 4, 5, and 6 for *We* = 11 are presented in Figure 9. It should be noted that, in contrast to Figure 8, data on the ordinate axis result in the maximum droplet spreading $D_m$, and on the abscissa axis, the moment of the droplet rebound from superhydrophobic surface is $t_b$ = 10.2 ms. This is carried out for a direct comparison with the results of Ding et al. [47], who investigated the liquid droplet spreading that falls on a silicon surface with a nanostructured microcone. The authors note that the fluid flowing into the structure leads to a fundamental difference in the behavior of the droplet on the retraction stage in comparison with a smooth surface, which is observed in our experiments. In the work of Ding et al. [47], the change in the contact angles was achieved either by changing the surface texture (a nanoporous structure on a microcone was created) or by mixing water with ethanol. The authors argue that the higher the static contact angle, the earlier the rebound from the superhydrophobic surface will occur, which is consistent with the results shown in Figure 8d. In addition, despite differences in liquid and structure types (Figure 9), the dynamics of water droplets spreading on our surfaces with different surface energies (but the same morphology) are identical to the results of Ding et al. [47] for similar contact angles. Our results in the range of *We* = 1.5–33 also agree very well with the work of M. Zhou [74], where the droplet ($D_0$ = 2 mm) falling on a carbon nanotube forest was considered, and the contact angle of the surfaces was varied by the plasma treatment intensity (Figure 8b,c and Figure 9b). Note that the data from this work generalize well using the dependence in Figure 5a, which gives a value of $t^*$ = 1.9 ms. Figure 9b shows that after $t/t^*$ = 1, the contact line dynamics on hydrophilic and superhydrophobic surfaces begin to differ. For comparison, Figure 8b also shows the water droplet spreading data on a smooth silicon surface with natural oxidation and it shows that the contact line dynamics is performed differently from the case of a droplet falling on a textured surface. A similar result is obtained for *We* ~ 33. Thus, it may be concluded that such a factor as the type of hierarchical structure has an insignificant effect on the falling droplets spreading on surfaces with high adhesion characteristics in the investigated range of *We* = 0.3–33, at least if the characteristic size of non-uniformities is less than 10 μm. However, on the other hand, according to Tang et al. [75], the change in texture roughness retained the character dynamics as spreading on a smooth surface, i.e., contact line fluctuations were observed (Wang 2020, Silicon in Figure 9).

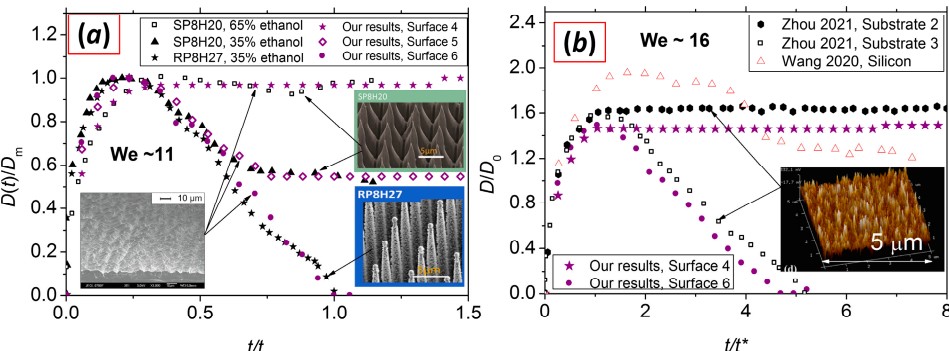

**Figure 9.** (**a**) Comparison with data of Ding et al. [47] for *We* = 11; the abscissa axis is normalized to the rebound moment $t_b$, the ordinate axis is normalized to the maximum spreading distance; (**b**) Comparison of our results for *We* = 15 with those of Zhou et al. [74] for *We* = 17.7 and Wang et al. [32] for *We* = 17. The insets show the microstructure of our surface and the surfaces from [32] and [74].

## 5. Conclusions

1.  For the first time, we systematically studied the dynamics of falling water droplets on surfaces with identical hierarchical structures but different wettability in a wide range of contact angles 5–161° for *We* = 0.3–33.
2.  We proposed a generalizing parameter—the time value $t^* = 0.66\,D_0^{3/2}$—corresponding to the transition between inertial and viscous flow regimes. We compared the dynam-

ics of water droplets falling at different velocities and onto different surfaces. It was shown that the parameter $t^*$ does not depend on the *We* number in all investigated conditions.

3. Analyzing the velocity fields obtained by the LBM, it was found that the inertial spreading regime $<t^*$ corresponds to the moment of capillary-surface waves reaching the droplet apex for all surfaces in the considered conditions. The inertial-capillary number $t_c$ corresponds to the zeroing of velocity for the superhydrophobic surface. However, for superhydrophilic surfaces, $t_c$ has no physical meaning.

4. It was shown that surfaces with absolutely different hierarchical structures can provide the identity of the contact line dynamics for falling droplets, regardless of the liquid used, where the contact angles equality is the necessary condition.

5. It was found that the droplet spreading over surfaces with high adhesion force (or exhibiting the rose petal effect or, in other words, having very large contact angle hysteresis) is fundamentally different from droplets spreading over a smooth surface despite the equality of contact angles. For the first time, it was shown that the surface structure does not affect the dynamics of the falling droplet spreading if we deal with the rose petal effect, i.e., the key factor is the liquid to the surface adhesion force.

**Supplementary Materials:** The following supporting information can be downloaded at https://www.mdpi.com/article/10.3390/w15040719/s1, Figure S1: Snapshots of droplet $D_0$ = 2.3 mm falling on surface with different contact angle for We = 0.3, Figure S2: Snapshots of droplet $D_0$ = 2.3 mm falling on surface with different contact angle for We = 11, Figure S3: Snapshots of droplet $D_0$ = 2.3 mm falling on surface with different contact angle for We = 15, Figure S4: Snapshots of droplet $D_0$ = 2.3 mm falling on surface with different contact angle for We = 22, Figure S5: Snapshots of droplet $D_0$ = 2.3 mm falling on surface with different contact angle for We = 33.

**Author Contributions:** Conceptualization, M.-K.L.; methodology, V.T. and F.R.; validation, S.S.; investigation, N.M., A.R., and E.S.; resources, S.S. and A.S.; data curation, S.S. and A.S.; writing—original draft preparation, S.S.; writing—review and editing, S.S., F.R., and E.S.; visualization, N.M.; supervision, S.S.; project administration, S.S. All authors have read and agreed to the published version of the manuscript.

**Funding:** The reported study was funded by RFBR and NSFC, project number 21-52-53025 GFEN_a; the equipment for the study was provided as part of the financial support of a grant from the Government of the Russian Federation to support scientific research conducted under the guidance of leading scientists (mega-grant No. 075-15-2021-575).

**Data Availability Statement:** Not applicable.

**Conflicts of Interest:** The authors declare no conflict of interest.

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
