# Peer review of "Spreading of Impacting Water Droplet on Surface with Fixed Microstructure and Different Wetting from Superhydrophilicity to Superhydrophobicity"

_water, doi:10.3390/w15040719_

Round 1

Reviewer 1 Report

- It would be helpful if the authors could add more details at the conclusion of the introduction, especially expressing the paper's goal, which is already stated but requires a bit more clarification.

- Authors must update their literature review to include more current studies, such as those published in 2020-2022.

- The language of the document requires expert touch-ups because some portions of the work include typos and mistakes that must be minimized.

- Explain more about the dynamics of falling droplets spreading research background in the introduction.

- In the conclusion part, writers should concentrate on the study's results and key findings, keeping them concise because the additional explanation is provided in the results and discussion sections. The velocity fields at different moments are obtained using the calculations carried out by the lattice Boltzmann method. Particularly it was found that the inertial spreading regime corresponds to the moment of capillary-surface waves reaching droplet apex and also to the appearance of vertically-directed velocity component! This very general result needs to be supported from within the text with numerical calculations

- Why does the droplet fixing moment for surface 5 not depend on "We"?

- Results and discussion section is well explained, please try to look at the figures (figs 5 and 6) in this section they might need more explanation.

- In falling drops, why did you not observe any significant difference in droplet behavior depending on size?

- using graphical abstract can increase the quality of the work.

Author Response

Response to Reviewer 1 Comments

We very thanks dear Reviewer for fruitfull comments.

Point 1: It would be helpful if the authors could add more details at the conclusion of the introduction, especially expressing the paper's goal, which is already stated but requires a bit more clarification.

Response 1: We have changed the text of the introduction and tried to place accents for greater clarity. Thanks for recommendation

Point 2: Authors must update their literature review to include more current studies, such as those published in 2020-2022.

Response 2: Many thanks for the recommendation! We have added references [16-20], [27], [28], [47]

Point 3: The language of the document requires expert touch-ups because some portions of the work include typos and mistakes that must be minimized.

Response 3: We thank the reviewer for their attention. The language of the manuscript has been revised

Point 4: Explain more about the dynamics of falling droplets spreading research background in the introduction.

Response 4: We have added a paragraph to the introduction section.

Point 5: In the conclusion part, writers should concentrate on the study's results and key findings, keeping them concise because the additional explanation is provided in the results and discussion sections. The velocity fields at different moments are obtained using the calculations carried out by the lattice Boltzmann method. Particularly it was found that the inertial spreading regime corresponds to the moment of capillary-surface waves reaching droplet apex and also to the appearance of vertically-directed velocity component! This very general result needs to be supported from within the text with numerical calculations.

Response 5: We completely agree with these comments. The corresponding conclusion is reformulated in accordance with the part of the text in which it is discussed.

Point 6: Why does the droplet fixing moment for surface 5 not depend on "We"?

Response 6: This is a rather interesting result. In order to confirm it, we added a file with Supplementary Materials containing experimental results for the numbers We = 0.3-33, which were not included in the main article (similar to Figure 3). One can see that ~8 ms corresponds to the moment the droplet is fixed to the surface. The fixation of the droplet occurs as a result of the Cassie-Wenzel transition (which does not occur for the superhydrophobic surface 6 but is realized for surface 5). In other words, the fixation of the droplet is due to the displacement of air from the surface microstructure, which occurs at the initial stage of contact between the droplet and the surface. We believe that the kinetic energy in the droplets is spent to a greater extent on the work of overcoming the surface tension, and not on the displacement of air from the microstructure. It is known [1], the duration of contact of a droplet with a superhydrophobic surface does not depend on the We number, but only on its size. A similar scenario can be expected in the case of a droplet falling on an almost superhydrophobic surface 5. Thus, the area from which air will be displaced will be determined not by the We number, at least for the studied range, but by the droplet size. Consequently, the moment of fixation will also be independent of the number We. We are grateful to the referee for this question and text has been added after Figure 8.

  1. D. Richard, C. Clanet, D. Quéré, Contact time of a bouncing drop, Nature. 417 (2002) 811–811. doi:10.1038/417811a.

Point 7: Results and discussion section is well explained, please try to look at the figures (figs 5 and 6) in this section they might need more explanation.

Response 7: We have added explanations for figs 5 and 6 to the manuscript.

Point 8: In falling drops, why did you not observe any significant difference in droplet behavior depending on size?

Response 8: In our work, we used droplets significantly different in volume (several times). Indeed, if only the absolute size of the droplet changes, the spreading dynamics will change significantly. At the same time, our analysis is based on the theory of similarity, we believe that the Weber and Reynolds numbers in a dimensionless form (in addition to the contact angle) adequately describe the behavior of the droplet.

Point 9: Using graphical abstract can increase the quality of the work..

Response 9: Thanks for the advice - we have prepared the graphical abstract!.

Reviewer 2 Report

In this article, the authors investigated the behavior of water droplets falling on hydrophobic and hydrophilic surfaces with various contact angles ranging from 0 to 160°. They also made a qualitative comparison with LBM simulation results. The manuscript is well prepared and the research objectives are clearly explained in the results and discussion section. However, before accepting the manuscript for publication, the authors should modify the paper according to the comments mentioned below.

  • There are money typos and unnecessary left indentations or spaces for a paragraph start. The indentation space for all the equations should be the same. Please correct the manuscript for some typos and grammatical mistakes.

  • The introduction is missing the present study's novelty compared to the previous studies. The authors should add a clear discussion that states the purpose and implications of the study.

  • The qualitative comparison of the experimental results with LBM results is not satisfactory. Consider choosing a three-dimensional LBM model for this (or) Mention this point in the manuscript why the authors only worked with the 2D model.

  • I believe that the authors used Shan-Chen's [49] phase separation model, which is only suitable for small-density ratios, to deal with the droplet spreading problem. But, I recommend considering a high-density ratio multiphase model as the densities ratio is very high for the current study

  • I suggest including the simulation results also in Figure 7 as the readers may want to see them.

Author Response

We very thanks dear Reviewer for fruitfull comments.

Point 1: There are money typos and unnecessary left indentations or spaces for a paragraph start. The indentation space for all the equations should be the same. Please correct the manuscript for some typos and grammatical mistakes.

Response 1: We are very thankful for the reviewer's comment. The correction of the manuscript was done

Point 2: The introduction is missing the present study's novelty compared to the previous studies. The authors should add a clear discussion that states the purpose and implications of the study.

Response 2: The introduction section was worked out and the emphasis was revised. We are grateful for the comment.

Point 3: The qualitative comparison of the experimental results with LBM results is not satisfactory. Consider choosing a three-dimensional LBM model for this (or) Mention this point in the manuscript why the authors only worked with the 2D model.

Response 3: The reviewer rightly points out the significant assumptions that the two-dimensional formulation introduces. We added clarifications to the article. The essence of these explanations lies in the fact that the two-dimensional formulation quite adequately describes the physics of the process of interaction of a droplet with a surface. This is confirmed by many works in which such a statement was used. Also, note that the main purpose of the simulation was the need to obtain the distribution of the velocity vector inside the droplet, which was not available from the experimental results.

Point 4: I believe that the authors used Shan-Chen's [49] phase separation model, which is only suitable for small-density ratios, to deal with the droplet spreading problem. But, I recommend considering a high-density ratio multiphase model as the densities ratio is very high for the current study

Response 4: The Shang-Chen model in its original formulation [62] is indeed limited only by small ratios of the densities of the liquid and gas phases (~10). However, the modifications of this method developed in [64, 67-69], used in this work, make it possible to stably model the dynamics of two-phase flows with a density ratio of the order of hundreds-thousands (100-1000), we note that in [68] it is reported about successful modeling of a flat interface at a density ratio of ~109. We have somewhat expanded the description of the method to clarify this remark. Note, that references list in revised manuscrip was changed.

Point 5: I suggest including the simulation results also in Figure 7 as the readers may want to see them.

Response 5: Figure 7 is the result of the simulation. We believe that the reviewer had in mind Figure 8. This figure is a comparison of our data and other works, demonstrating the discrepancy in the dynamics of the contact line on smooth and textured surfaces. We would not like to overload this figure with calculated curves. However, at the recommendation of the reviewer, we have added the simulation results for the surface 4 and surface 5 cases to Figure 3a. The simulation, as expected, well describes the initial stage of droplet contact with the surface, however, at later moments a discrepancy is observed due to the 2D approach

Reviewer 3 Report

The paper is well written, the presented contents are well organized. Few comments are given for the authors' considerations, I believe that the corrected work will be available for publication.

1. The description of research objectives in Introduction is not clear enough.

2. It is suggested to introduce the experimental method and the simulation method in more detail.

3. If there are other impurities in the water droplet, how might that affect the results of the experiment?

4. Some methods in these references about two-phase flow is helpful to improve this article, such as:

Entropy and exergy analysis of the steam jet condensation in crossflow of subcooled water. Annals of Nuclear Energy, 2023, 180: 109485.

Droplet impact onto moving liquids, JOURNAL OF FLUID MECHANICS, 809 , pp.716-725

Signal selection for identification of multiphase flow patterns in offshore pipeline-riser system. Ocean Engineering, 2023, 268: 113395.

A distributed analytical electro-thermal model for pouch-type lithium-ion batteries. Journal of the electrochemical society, 2014, 161(14): A1953.

it is suggested to reference these articles.

5. How might coatings of different kinds of substances affect the results?

Author Response

Response to Reviewer 3 Comments

We very thanks dear Reviewer for fruitfull comments.

Point 1: The description of research objectives in Introduction is not clear enough..

Response 1: The introduction section was worked out and the emphasis was revised. We are grateful for the comment

Point 2: It is suggested to introduce the experimental method and the simulation method in more detail.

Response 2: We added information about the laser scanning process in the experimental section. Also, we included data about conditions in experiments of fluoropolymer deposition. The detail of contact angle measurements was introduced. We have also expanded the description of calculation methods.

Point 3: If there are other impurities in the water droplet, how might that affect the results of the experiment?

Response 3: The properties of the used fluid dramatically influence the behavior of droplets. Impurities in the water droplet, in particular, the presence of surfactants can significantly affect the surface tension of the droplet, as well as the contact angle. That's why we used Milli-Q water, which is very pure distilled and nano-filtered water. We also measured the surface tension of Milli-Q water on a KRUSS tensiometer, which coincided with the table values (72 mN/m). We have added information about the type of liquid used in the text of the article

Point 4: Some methods in these references about two-phase flow is helpful to improve this article, such as:

Entropy and exergy analysis of the steam jet condensation in crossflow of subcooled water. Annals of Nuclear Energy, 2023, 180: 109485.

Droplet impact onto moving liquids, JOURNAL OF FLUID MECHANICS, 809 , pp.716-725

Signal selection for identification of multiphase flow patterns in offshore pipeline-riser system. Ocean Engineering, 2023, 268: 113395.

A distributed analytical electro-thermal model for pouch-type lithium-ion batteries. Journal of the electrochemical society, 2014, 161(14): A1953.

it is suggested to reference these articles.

Response 4: We are grateful for the recommendation of a respected reviewer! The references [27-29] were added to the manuscript.

Point 5: How might coatings of different kinds of substances affect the results?

Response 5: This is a rather important question. According to the data available in the literature, very different approaches are used to change the wetting properties. For example, Boinovich’s group uses homemade complex water hydrophobic agents [1]. Other groups use water repellents such as silanes or FAS [2-4]. Such functionalizing layers are quite thin, on the order of one or two monolayers, and allow one to achieve water pronounced repellent effect. In other words, even for very thin layers, the effect of superhydrophobicity is achieved. Upon contact with such a surface, the droplet bounces off even at small We numbers. Our experience shows that when using FAS and a fluoropolymer coating, the rebound dynamics are almost identical. Small discrepancies in the dynamics, for example, a difference in the time of contact with the surface up to 10%, can be associated with the peculiarities of the transition between the wetting states, especially at high We.

Many other teams use commercial water repellents, such as NeverWet [5], whose thickness is uneven and reached up to several hundred micrometers, i.e. completely changing the topology of the used material. In this case, superhydrophobic states will also be achieved, however, the topology developed on such large scales can affect many aspects when a liquid spreads over a surface. This is confirmed by the work of Antonin [6], which shows that the morphology, or rather, the change in its surface roughness, affects the receding contact angle and, accordingly, the droplet rebound time. From this point of view, the paper we propose is interesting in that we fix the topology of the surface and keep the values of the receding contact angle close

[1] L.B. Boinovich, A.M. Emelyanenko, The behaviour of fluoro- and hydrocarbon surfactants used for fabrication of superhydrophobic coatings at solid/water interface, Colloids Surfaces A Physicochem. Eng. Asp. 481 (2015) 167–175. doi:10.1016/j.colsurfa.2015.05.003.

[2]          W. Ding, C.A. Dorao, M. Fernandino, Improving superamphiphobicity by mimicking tree-branch topography, J. Colloid Interface Sci. 611 (2022) 118–128. doi:10.1016/j.jcis.2021.12.056.

[3]          Y. Qing, C. Yang, Q. Zhao, C. Hu, C. Liu, Simple fabrication of superhydrophobic / superoleophobic surfaces on copper substrate by two-step method, J. Alloys Compd. 695 (2017) 1878–1883. doi:10.1016/j.jallcom.2016.10.323.

[4]          H. Zhu, Z. Guo, Understanding the Separations of Oil/Water Mixtures from Immiscible to Emulsions on Super-wettable Surfaces, J. Bionic Eng. 13 (2016) 1–29. doi:10.1016/S1672-6529(14)60156-6.

[5]          F. Wang, T. Fang, Retraction dynamics of water droplets after impacting upon solid surfaces from hydrophilic to superhydrophobic, Phys. Rev. Fluids. 5 (2020) 033604. doi:10.1103/PhysRevFluids.5.033604.

[6]          C. Antonini, F. Villa, I. Bernagozzi, A. Amirfazli, M. Marengo, Drop rebound after impact: The role of the receding contact angle, Langmuir. 29 (2013) 16045–16050. doi:10.1021/la4012372.

Round 2

Reviewer 1 Report

The comments of my first report have been addressed by the authors.

Reviewer 2 Report

The manuscript can be accepted in the present form